# Induced Cardiomyocyte Proliferation: A Promising Approach to Cure Heart Failure

**DOI:** 10.3390/ijms22147720

**Published:** 2021-07-19

**Authors:** Abou Bakr M. Salama, Ahmad Gebreil, Tamer M. A. Mohamed, Riham R. E. Abouleisa

**Affiliations:** 1Department of Medicine, Division of Cardiovascular Medicine, Institute of Molecular Cardiology, University of Louisville, 580 S. Preston St., Rm 122, Louisville, KY 40202, USA; aboubakr.salama@louisville.edu (A.B.M.S.); ahmed.gibreil@louisville.edu (A.G.); tamer.mohamed@louisville.edu (T.M.A.M.); 2Department of Cardiovascular Medicine, Faculty of Medicine, Zagazig University, Zagazig 44519, Egypt; 3Department of Cardiac Surgery, Verona University, 37134 Verona, Italy; 4Diabetes and Obesity Center, Department of Medicine, Division of Environmental Medicine, Christina Lee Brown Envirome Institute, University of Louisville, Louisville, KY 40202, USA; 5Department of Pharmacology and Toxicology, University of Louisville, Louisville, KY 40202, USA; 6Institute of Cardiovascular Sciences, University of Manchester, Manchester M13 9PL, UK; 7Department of Biochemistry, Faculty of Pharmacy, Zagazig University, Zagazig 44519, Egypt

**Keywords:** cardiomyocyte proliferation, cardiac regeneration, cell cycle

## Abstract

Unlike some lower vertebrates which can completely regenerate their heart, the human heart is a terminally differentiated organ. Cardiomyocytes lost during cardiac injury and heart failure cannot be replaced due to their limited proliferative capacity. Therefore, cardiac injury generally leads to progressive failure. Here, we summarize the latest progress in research on methods to induce cardiomyocyte cell cycle entry and heart repair through the alteration of cardiomyocyte plasticity, which is emerging as an effective strategy to compensate for the loss of functional cardiomyocytes and improve the impaired heart functions.

## 1. Introduction

Heart failure is the most prominent cause of hospitalization and one of the leading causes of mortality globally. During heart failure progression, damaged cardiomyocytes are replaced by fibrotic tissue, since cardiomyocytes are not able to regenerate themselves when damaged. This subsequently triggers cardiac remodeling and heart failure [1,2]. All the current treatment strategies for heart failure are symptomatic treatments to slow down its progression. Hence, heart transplants, which are eminently scarce and not without complications, are as yet the only cure for heart failure [3].

Spontaneous cardiomyocyte regeneration has been demonstrated in embryonic and neonatal mammals after genetic ablation, apical resection, or myocardial infarction [4,5,6]. Adult cardiomyocyte proliferation and turnover have been reported to be minimal in human hearts [7,8] and rodents [9,10]. Cardiomyocyte division was also demonstrated to occur at a very low rate after acute and chronic infarction in humans [11]. These findings projected a view of the heart as a dynamic organ constituted of myocytes that may regenerate if prompted. However, knowledge on the molecular mechanisms underpinning cardiomyocyte regeneration is still primitive compared to that for other cell types. This review aims to summarize the recent progress relating to the methods and targets used to promote adult cardiomyocyte cell cycle entry and highlights new possibilities for the enhancement of cardiomyocyte regenerative capacity that might lead to new effective therapeutic approaches for heart failure.

## 2. Induction of Cardiomyocyte Cell Cycle Entry through Direct Cell Cycle Activation

Overexpression of cell cycle activators or knocking down of cell cycle inhibitors have been widely used to induce cardiomyocyte cell cycle entry. The cyclin D family and its related kinases (CDKs) are known to regulate the transition of the cell cycle from G1 to S phase, induce DNA synthesis, and maintain cell cycle activity [12,13,14]. Cyclins A, D, and E, as well as the related kinases, are also involved in DNA synthesis, while cyclin B is crucial for cytokinesis [15]. It has been shown that cyclin A and B disappear in adult hearts, while cyclin D, E, CDK2, and CDK4 expression and activity are very low in adult hearts [16,17].

In adult mammalian cardiomyocytes, the complexes of cyclins/CDKs are constitutively inhibited by two families: the CIP/KIP family, including P21, P27, and P57; and the INK4 family, including p16, p15, p18, and p19. In general, upregulation of the CIP/KIP family and INK4 family arrests the cell cycle by blocking CDKs’ regulation of G1/S and G2/M checkpoints [16,18,19]. The CIP/KIP family members P21 and P27 block all phases of the cell cycle, forcing the postnatal cardiomyocytes to exit the cell cycle. On the other hand, P57 is expressed during mid-gestation and it plays a role in cardiomyocyte differentiation, while its increased expression in adult cardiomyocytes is associated with lower proliferation capacity and more differentiation [20,21,22]. The INK4 family members activate responsive retinoblastoma (Rb) proteins to maintain a quiescent state and induce a hypertrophic response in adult cardiomyocytes [16,23]. The P38 mitogen-activated protein kinases are also known to inhibit the cell cycle and induce cardiomyocyte differentiation by decreasing cellular expression of cyclin A2, cyclin B, and cdc2a [24,25]. Many studies have targeted one or more cell cycle gene regulators to induce cardiomyocyte cell cycle entry in trials to regenerate hearts after injury (Table 1). Mice with cardiac-specific cyclin D1 overexpression showed a concomitant increase in CDK4 in adult myocardium, with a modest increase in CDK2 and proliferating cell nuclear antigen. Histochemical analysis showed an abnormal pattern of multinucleation and sustained DNA synthesis [26]. Cyclin G1 expression was also linked to induction of S phase with the arrest of cytokinesis, leading to polyploidy and multinucleation in cardiomyocytes [27]. In another study, Tane and his colleagues showed similar findings as they investigated the cause of multinucleation and arrest of mitosis. They showed that the cardiomyocytes of transgenic mice overexpressing cyclin D only fail to increase the level of Cdc25 (a and b) by the end of the G2 phase. Cdc25 (a and b) is a cyclin of the specific phosphatase family that activates the cyclin B1/CDK1 complex in cells for entry into mitosis [28]. Pasumamarthi et al. compared the effect of overexpressing cyclins D1, D2, and D3 on cell cycle induction in vivo. They showed persistent cell cycle activity and infarction regression after cardiac injury in adult cyclin D2 transgenic mice, while cyclin D1 and D3 transgenic mice failed to show the same phenotype [13].

Cyclin A2 is expressed during embryonic development of the heart and the early postnatal period and silenced afterward [16]. Many studies have investigated the effect of cyclin A2 expression on cell cycle resumption. Chaudhry et al. showed cardiac enlargement with hyperplasia of cardiomyocytes in adult transgenic mice persistently expressing cyclin A2 [29]. Woo et al. used an adenoviral vector to deliver cyclin A2 to mice hearts following left anterior descending (LAD) artery ligation. Cyclin A2-treated rats showed cardiac cell cycle activation associated with an improvement in hemodynamics, preload-adjusted maximal power, and cardiac output compared to control-treated rats [30]. Similar findings were reported by Cheng et al. in cyclin A2 overexpression transgenic mice subjected to LAD artery ligation [31]. Shapiro et al. confirmed these findings in large animals, namely pigs. Cyclin A2-treated pigs showed an ~18% increase in ejection fraction compared to control-treated pigs. Cultured porcine adult cardiomyocytes showed a more than threefold increase in the phospho-histone H3 (PHH3) in cyclin A2-expressing animals compared to controls [32].

Multiple combinations of different cell cycle genes have also been investigated for more efficient induction of the cell cycle. Tamamori-Adachi et al. hypothesized that nuclear localization of cyclin D1 is required for induction of the cell cycle in postnatal cardiomyocytes. Accordingly, they promoted cell cycle re-entry by using a combination of adenoviruses expressing modified cyclin D1 with a nuclear localization signal (NLS) and CDK4. Isolated postnatal cardiomyocytes treated with this combination showed a three- to fivefold increase in expression of Ki-67 and BrdU [33]. Katrina et al. used a combination of cyclin B1 and constitutively active cell division cycle 2 kinase (CDC2AF) to induce cell cycling in fetal, neonatal, and adult cardiomyocytes isolated from Wistar rats and showed a significant increase in the number of tropomyosin-positive cells [34].

Mohamed et al. showed efficient induction of stable cell cycle division, assessed by EdU and PHH3, in 15–20% of adult mouse, rat, and hiPS cardiomyocytes overexpressing a combination of four cell cycle genes; cyclin-dependent kinase 1 (CDK1), CDK4, cyclin B1, and cyclin D1 (termed 4F). Cell cycle induction using 4F also showed robust complete cytokinesis in vivo using MADM mice [35]. In a trial for preclinical application of 4F, Abouleisa et al. cloned a polycistronic non-integrating lentivirus encoding 4F in which each factor was driven by TNNT2 promotor (TNNT2-4F-NIL) to induce cell cycling transiently and specifically in cardiomyocytes. Intramyocardial injection of TNNT2-4F-NIL improved systolic functions and reduced the scar size after ischemic reperfusion in rats and pigs compared to control-treated rats and pigs [36].

Another idea for the activation of cell cycle genes is to target cell cycle inhibitors. Sirt-1 is a member of the histone deacetylase (HDAC) family, which deacetylate multiple targets including P21 protein. Hence, the deacetylated form of P21 is less stable and more prone to ubiquitination. Sirt-1 overexpression showed a significant reduction in active P21 acetylation, which resulted in a release of the inhibitory effect on the cell cycle genes and a significant increase in the mitotic markers, including PHH3 and Aurora B kinase-positive nuclei [37].

A combination of P21, P27, and P57 siRNAs results in the progression of neonatal and adult cardiomyocytes to the S phase, indicated by an increase in the ratio of the number of nuclei to cardiomyocytes. In one study, increases in the total number of cardiomyocytes suggested that a subpopulation can go beyond karyokinesis [23]. Similarly, Engel et al. have shown that cardiac-specific P38α knockout mice exhibited an increase in cardiomyocyte mitosis, indicated by increased BrdU and pHH3-positive nuclei, and that this was associated with an increase in the cell counts in vitro and in vivo [24].

## 3. Transcription Factors Control Cardiomyocyte Cell Cycle Re-Entry

Transcription factors have been shown to govern postnatal cell cycle arrest in adult cardiomyocytes, as summarized below and in Table 2.

**E2F** is a transcription factor family that activates genes responsible for DNA replication and cell cycle progression. The E2F family comprises eight genes encoding nine proteins (E2F1, E2F2, E2F3a, E2F3b, E2F4, E2F5,E2F6, E2F7, and E2F8) [38]. E2F4 is a transcription factor that is required for cardiomyocyte cell cycle entry. During mouse heart development, E2F4 nuclear expression is reduced in association with the loss of cardiomyocyte proliferation capacity. FGF1/p38i-stimulated neonatal cardiomyocytes exhibit an increase in nuclear E2F4 expression. E2F4 knockdown significantly reduced the number of PHH3-positive cardiomyocyte nuclei after FGF1/p38i, although it did not affect the BrdU-positive cardiomyocyte nuclei, suggesting that E2F4 is required during the G2/M phase. E2F4 induces cyclin A and cyclin E expression [39,40]. E2F2 overexpression significantly increased the number of BrdU, PHH3, and Aurora kinase-positive cardiomyocyte nuclei [39,40]. Rb, a regulatory pocket protein of the E2F complex, combines with E2F2. CDK2 and CDK4 phosphorylate Rb protein, which results in its release from E2F transcription factors. Active E2F activates the expression of genes responsible for DNA synthesis and cell cycle induction [41,42]. Although these studies showed the importance of the E2F family in inducing cardiomyocyte cell cycle entry, other studies have proved that the expression of E2F1, E2F3, and E2F5 leads to apoptosis [43,44].

The **Meis** transcription factor family is part of the three amino acid loop extension (TALE) family of homeodomain transcription factors. Meis1 is an essential regulator for normal embryonic hematopoiesis, heart development, and stem cell (HSC) quiescence [45,46,47,48]. Although Meis1 plays a central role in cardiac differentiation during development, the role of Meis1 in adult cardiomyocytes is unclear [49,50]. A recent study demonstrated that Meis1 imposed a limit on the postnatal proliferation capacity of cardiomyocytes. Meis1 is highly expressed in P7 cardiomyocytes, which is correlated to postnatal cell cycle arrest. Meis1 KO mice extend the postnatal proliferation capacity of cardiomyocytes to 14 days. Meis1 KO showed an increase in the proliferation markers in adult cardiomyocytes, with no deleterious effect on cardiac function or induction of hypertrophy. Mechanistically, Meis1 is a transcription factor that activates cell cycle inhibitors P15, P16, and P21, which inhibit cell cycle genes CDK6, -4, and -2 and decrease cyclin D and E [51].

**The T-box (Tbx)** gene family contains transcriptional regulators that play a crucial role in regulating many aspects of embryogenesis in a wide variety of organisms [52]. The complicated structure of the heart develops from a simple tube by polar elongation, myocardial differentiation, and morphogenesis. The role of Tbx family members during cardiac development has been identified [53]. Tbx20, a member of the Tbx-1 superfamily, is a key regulator for cardiomyocyte proliferation during development. Tbx20 deletion results in embryogenic death during mid-gestation [53]. Tbx20-induced specific overexpression in adult cardiomyocytes promoted cardiomyocyte cell cycle entry at the baseline and after MI. Tbx20-specific overexpression in adult cardiomyocytes improved cardiac function, scar size, and survival rate 4 weeks post-MI due to increased cell cycle re-entry [54,55]. Adult cardiomyocytes overexpressing Tbx20 have fetal-like characteristics, including mononucleated cardiomyocytes, upregulation of fetal contractile protein expression, and upregulation of the signaling pathways involved in cell cycle re-entry, such as P13K/AKT, Hippo/YAP, and BMP/Smad1/5/8. Tbx20 also represses the expression of cell cycle inhibitors P21, Meis1, and Btg2 [54,55]. Cardiac-specific Tbx20 KO in adult mice resulted in the onset of severe cardiomyopathy accompanied by arrhythmias, with death ensuing within 1 to 2 weeks of Tbx20 ablation [56].

**Gata4** is a member of the GATA family of transcription factors. GATA4 is a key regulator of cardiac development [57,58]. GATA4 is highly expressed in embryogenic and early neonatal cardiomyocytes and its expression is downregulated upon postnatal cell cycle arrest. Cardiomyocyte GATA4 is essential for heart regeneration. Overexpression of GATA4 increases cardiomyocyte proliferation after P7 cryoinjury while GATA4 cardiac-specific KO mice showed a larger scar size, reduced angiogenesis, and cardiomyocyte cell cycle entry post-P0 cryoinjury [59].

**Homeobox (HOX)** transcription factors play a key role in many aspects of cellular physiology, embryonic development, and tissue homeostasis. HOX13 expression is upregulated in p7 cardiomyocytes like Meis1 and reveals the same phenotype. A combination of Meis1–Hoxb13 double-knockout in adult hearts showed widespread cardiomyocyte cell cycle entry compared to single KO and improved left ventricular systolic function following MI [60].

## 4. MicroRNA Regulates Cardiomyocyte Cell Cycle Entry

MicroRNAs consist of single-stranded RNA which play several roles in regulating gene expression during cardiomyocyte development [61] (Table 3).

The **miR-302-367** cluster is expressed in the early developmental stage in mouse hearts and it is involved in modulating cardiomyocyte proliferation during the embryonic stage. Expression of miR-302-367 led to continuous cardiomyocyte proliferation and reduction of scar size after MI, probably through inhibition of the Hippo signaling pathway. However, it also led to immature cardiomyocyte dedifferentiation and heart failure, while transient expression of miR-302-367 results in more controlled proliferation and does not deteriorate cardiac function [62].

**miR-199a** is one of the microRNAs that is most effective in inducing robust cardiomyocyte cell cycle entry in neonatal and adult animals. The animal overexpressing miR-199a showed almost complete recovery of cardiac function parameters [63]. However, prolonged expression of miR-199a in larger animals results in ventricular arrhythmia [64].

**miR-128** is a microRNA the upregulation of which coincides with cell cycle arrest. Neonatal mice overexpressing miR-128 showed impaired function and attenuated proliferation capacity, while miR-128 KO mice showed a prolonged postnatal proliferation window through suppressing CDK inhibitor p27 and activation of cell cycle regulator cyclin E and of CDK2. Adult KO mice showed improved scar size and cardiac function after MI [65]. In this study, the authors suggested that the reduction in scar size was attributed to the dedifferentiation of the adult cardiomyocytes and their ability to proliferate, hence preventing the development of fibrosis.

**miR-294** is another microRNA that is expressed during prenatal development and its expression is lost in adult cardiomyocytes. miR-294 expression in neonatal myocytes showed a significant increase in cell cycle markers Ki69, PHH3, and Aurora kinase-positive nuclei compared to control neonatal myocytes. miR-294–treated mice showed a significant improvement in scar size and apoptosis after MI compared to control mice. miR-294 increased cardiomyocyte proliferation through repression of Wee1 activity, which led to the activation of the cyclin B1/CDK1 complex [66].

Furthermore, another microRNA, **miR-499**, has been found to regulate cardiomyocyte cell cycle entry and apoptosis via modulation of Sox6 and cyclin D1 activity [67].

## 5. Signaling Pathways That Control Cardiomyocyte Proliferation

Many signaling pathways interact to balance the progression and arrest of the cell cycle and apoptosis in mammalian cells [70]. Several pathways interact with cyclin-dependent kinases inhibitors (CDKIs) or directly with cyclins and CDKs to activate or inhibit cell cycle progression. Understanding the basic bio-physiology of these signaling pathways can provide a tool for controlled induction of cardiomyocyte proliferation in the hope for a regenerative niche [70].

**PI3K-AKT** is one of the major signaling pathways controlling the cell cycle. Increased AKT activity is associated with a longer half-life of cyclin D, increased CDK2 and CDK7 levels, and decreased expression of cell cycle inhibitors P21/P27/P57, which in turn promotes cell cycle progression [71,72,73]. Studies of small molecules activating the PI3K-AKT signaling pathway showed a successful induction of mitosis and cytokinesis in neonatal and adult cardiomyocytes [74]. PI3K-AKT interacts with two other fundamental pathways to facilitate progression into the cell cycle: the Hippo signaling pathway and Wnt/β-catenin [75]. Interestingly, the PI3K-AKT pathway could act as a survival pathway [76,77], which might imply that the beneficial effects post-infarction represent a salvaging of cell death in addition to the induction of the cardiomyocyte cell cycle.

**The Hippo signaling pathway** regulates cell proliferation mainly through phosphorylation/dephosphorylation of the transcriptional co-activator yes-associated protein (YAP). YAP is known to be required for normal fetal heart growth and it regulates insulin-like growth factor-1 (IGF-1) in developing hearts [78,79]. YAP interacts directly with the Myb–MuvB (MMB) complex to facilitate G2/M transition [80]. Monroe et al. showed that an active version of the Hippo pathway effector YAP (YAP5SA) can program adult mice cardiomyocytes to a fetal heart program with high proliferation capacity and increased chromatin accessibility [81]. Other studies have shown that YAP activation or overexpression in mice hearts enhances cardiomyocyte proliferation and improves outcomes after myocardial infarction. Xin et al. showed that cardiac-specific deletion of YAP in mice using α-myosin heavy chain C (αMHC-c)-Cre was associated with impaired cardiac functions with ventricular wall fibrosis, development of dilated cardiomyopathy, and death by 9 weeks of age. LAD artery ligation in these P2 day neonatal mice showed limited regenerative capacity compared to controls. Transgenic mice overexpressing a constitutively active YAP showed enhanced proliferative capacity and increased PHH3 expression after LAD artery ligation in P7 day old neonatal mice [82]. Lin et al. reported similar findings after MI induction in adult mouse hearts and YAP overexpression using adenoviral vector [83].

The **Notch signaling** pathway controls cellular proliferation and trabeculation in developing hearts [84]. Notch signaling activates bone morphogenic protein-10 (BMP-10), a protein involved in cardiomyocyte growth and maturation during development [84,85]. BMP10 has been shown to increase myocyte cell cycle entry by downregulation of cell cycle inhibitors P21, P27, and P75 [85]. Notch signaling indirectly increases neuregulin expression. Neuregulin is a paracrine agonist of the ErbB1-4 tyrosine kinase receptor which stimulates intracellular PI3K-AKT signaling [84,86]. Furthermore, another study demonstrated the direct interaction between Notch signaling and the PI3K-AKT survival pathway through c-Met-mediated activation of AKT [87]. Zhao et al. showed that suppression of Notch signaling in zebrafish impairs their capacity to regenerate their heart after ventricular resection [86].

**The Wnt/β-catenin** signaling pathway is known to be active during cardiac development and during adult left ventricular remodeling [88]. In normal adult hearts, Wnt is under continuous negative regulation by upstream soluble frizzled-related proteins (sFRPs). Cardiac injury—for example, after MI—removes the inhibitory effect of sFRPs, allowing Wnt to bind to frizzled receptors and activate the complex canonical pathway leading to increased expression and nuclear localization of the transcription factor β-catenin, as well as non-canonical downstream pathways [88]. sFRP2 protein knockout mice showed reduced fibrosis and improved cardiac functions after experimental MI. The mechanisms underlying these findings are complex and wnt/β-catenin can be involved [89]. The Wnt/β-catenin pathway is also subject to an inhibitory effect of GSK3β, which markedly increases during the postnatal period and is thought to be involved in the postnatal cell cycle exit. GSK3β inhibition or knockout activates Wnt/β-catenin, which increases cyclin D and allows the cardiomyocytes to proceed into S phase [90,91].

**The Jak/Stat** signaling pathway is a novel and promising pathway for cell cycle induction. Cytokines binding to their receptors lead to dimerization of interleukin-6 signal transducer (IL6st), which activates Jak1. In return, Jak1 phosphorylates IL6st to form a docking site for Stat3 phosphorylation and activation. Active Stat3 translocate to the nucleus to stimulate the expression of several transcription factors involved in cell proliferation [92]. A study of myocarditis models showed that Stat3 facilitates cell cycle re-entry, while its inhibition is associated with limited ability of cardiomyocytes to proliferate and increased scarring in response to injury [92,93].

**The Hedgehog****(Hh)** signaling pathway is currently emerging as a promising target for cell cycle modulation. Hedgehog proteins are known to interact with specific transmembrane receptor proteins, named Patched receptors, which in turn regulates the activity of the Glioma-associated oncogene homolog (Gli) family of transcription factors [94]. Hedgehog is known to be active during embryogenesis, controlling cellular proliferation and differentiation [95]. Studies of cardiac regeneration in lower vertebrates, including zebrafish and newts, showed a role for Hh signaling in cardiomyocytes proliferation [96,97,98]. Sonic Hedgehog (SHh) protein reconstitutes the embryonic signaling pathways, so it has been introduced to promote cardiomyocyte cell cycle re-entry [99,100,101]. Mechanistically, Hh signaling stimulates the expression of cyclin D2 and cyclin E1 and inhibits P27 through Gli1–Mycn network interaction [102].

Two studies tried to investigate the relationship between **autonomic innervation** and cardiomyocyte proliferation. White et al. have shown that sympathetic innervation is required to maintain neonatal mammalian heart regenerative capacity. Chemical sympathectomy in P2 neonatal mice hearts significantly inhibited cardiac regeneration after apical resection [103]. While Liu et al. showed that β-adrenergic receptor (β-AR) stimulation in patients with tetralogy of Fallot and pulmonary stimulation is linked to cardiomyocyte cell cycle arrest through repression of the cytokinesis gene epithelial cell transforming 2 (ECT2). Inactivation of β-AR genes in mouse models, and β-AR blocking using propranolol in both mouse models and human subjects with tetralogy of Fallot, increased cardiomyocyte proliferative capacity [104]. Other studies on zebrafish and neonatal mice hearts showed that both pharmacological inhibitions of cholinergic nerve functions and mechanical denervation diminished cardiomyocyte proliferation. Cholinergic stimulation by carbachol extended the postnatal cardiomyocyte regeneration window [105]. In brief, a balanced autonomic drive of the heart is required to maintain the proliferative capacity of cardiomyocytes.

## 6. Metabolic Influence on Cardiomyocyte Division

**Hypoxia:** It has been suggested that hypoxia plays a role in cardiomyocyte proliferation and cell cycle arrest. Induction of systemic hypoxemia attenuates oxidative DNA damage and induces cardiomyocyte proliferation in adult mammals. Mice subjected to hypoxemia one week after induction of myocardial infarction showed a robust regenerative response with decreased myocardial fibrosis and improvement of left ventricular systolic function [106]. HIF1-alpha is a transcription factor complex known to regulate the cellular response to hypoxia. A recent study showed that there is a special cardiomyocyte population that is hypoxic during mid-gestation, which promotes cardiomyocyte proliferation via HIF1alpha [107]. Kimura et al. have also suggested that the presence of hypoxic cardiomyocytes contributes to the formation of new cardiomyocytes, which supports the idea that hypoxia signaling is a hallmark of cycling cardiomyocytes [9].

## 7. Metabolic Substrates

The regenerative capacity of cardiomyocytes is lost one week after birth, which is coincident with the switch of metabolism from anaerobic glycolysis to mitochondria oxidative phosphorylation (fatty acid oxidation) [108]. Several studies have highlighted the importance of cardiomyocyte metabolism during cardiomyocyte development and proliferation (Table 4). Cardoso et al.’s (2020) study revealed the role of mitochondrial substrate utilization in regulating cell cycle progression. Neonatal mice fed fatty acid-deficient milk showed extended cell cycle arrest for 14 days. Cardiac-specific pyruvate dehydrogenase kinase 4 (PDK4) KO mice showed an increase of glucose compared to fatty acid utilization, which increased cardiomyocyte proliferation. These data suggest that fatty acid utilization contributes to cell cycle arrest in cardiomyocytes [109].

Other studies have tried to elevate the role of glycolysis in regulating cardiomyocyte cell cycle re-entry (Table 5). Two studies suggested the involvement of pyruvate kinase muscle isoenzyme 2 (**Pkm2**), an isoenzyme of the glycolytic enzyme pyruvate kinase, in regulating cardiomyocyte cell cycle re-entry. Magadum et al. (2020) showed that Pkm2 is expressed in cardiomyocytes during development and immediately after birth. Its expression is lost in adult cardiomyocytes. Pkm2 cardiomyocyte-specific KO mice showed a significant reduction in the number of cardiomyocytes, a significant increase in the size of myocytes, and a lower proliferation capacity, while cardiomyocyte-specific Pkm2-modified RNA overexpression resulted in a significant increase in proliferation markers, improved cardiac function, and reduced scar size after MI [110]. Conversely, another study by Hauck et al. (2020) showed that induced cardiac-specific Pkm2 KO promotes cardiomyocyte cell cycle re-entry, improves cardiac function, and reduces scar size after MI [111]. Both of the studies referred to the interaction between Pkm2 and B-catenine (anabolic pathway) in the regulation of cardiomyocyte cell cycle entry [110,111].

In addition, a recent study demonstrated that the metabolic switch from oxidative phosphorylation to glycolysis and lactate fermentation modulates adult cardiomyocyte cell cycle re-entry. Cardiomyocyte-specific overexpression of a constitutively active ErbB2 receptor (caErbB2 OE) upregulates genes involved in glycolysis (e.g., Pfkp, Pdk3, and Pkm2), as well as glucose and lactate transporters (Slc16A3 and Slc2A1), and downregulates genes transcribed from mitochondrial DNA. These metabolic changes have been shown to drive cardiomyocyte cell cycle re-entry. These results are in line with other findings that showed the Notch signaling pathway driving cardiomyocyte proliferation [112,113].

A recent study highlighted the role of succinate in regulating cardiomyocyte proliferation. Injection of malonate, an inhibitor of succinate dehydrogenase (SDH), extends the proliferation window of neonatal cardiomyocytes to 14 days. Interestingly continuous injection of malonate after MI promotes adult cardiomyocyte cell cycle re-entry, revascularization, and heart regeneration via metabolic reprogramming [114].

Moreover, a novel drug screening of hiPS cardiac organoids revealed the role of the mevalonate pathway in regulating cardiomyocyte proliferation [115]. The mevalonate pathway is an anabolic pathway that converts acetyl-CoA into isopentenyl pyrophosphate, the essential building block of all isoprenoids [116].

## 8. Recent Genetic Tools to Track Cardiomyocyte Cell Cycling

Cardiomyocytes can activate cell cycling without complete cytokinesis and, in many cases, cardiomyocytes may undergo bi- (or multi-) nucleation, endoreplication, polyploidization, or cellular hypertrophy [117,118,119,120]. Therefore, several recent studies have aimed to develop or use existing transgenic lineage tracing models to distinguish between activation of the cell cycle and true cytokinesis in vitro and in vivo. Mosaic analysis of the dual marker mouse model (MADM) [121] has become the gold standard to demonstrate true cytokinesis in cardiomyocytes in vivo, as demonstrated by several recent studies [35,36,60,109,122]. Several other interesting models have been recently generated to demonstrate the progress of cardiomyocytes through the cell cycle in vivo, such as the double transgenic Myh6-eGFP-anillin/Myh6-H2BmCh mice [123]. However, this mouse model does not permanently label the dividing cardiomyocytes and it requires live cell imaging of the heart slices, which is challenging. Another example is the use of the fluorescence ubiquitination cell cycle indicator (FUCCI) [124] to track the cell cycle progress in cardiomyocytes in vitro and in vivo. Interestingly, the use of the FUCCI cell cycle reporter revealed that small molecule regulators of PPARd signaling could modulate the cardiomyocyte cell cycle [125]. Furthermore, in vivo models [126,127] and hiPS-derived cardiomyocytes [128] have been generated to study cell cycle progression in the heart using FUCCI indicators. Interestingly, a recent study generated a Ki67-based lineage and traced mice to track the cell cycle activation in cardiomyocytes [129]. Finally, a recent preprint described the generation of an Aurora kinase B-based reporter to be used in identification of cardiomyocyte cell cycle activation in vitro and in vivo [36]. These new tools will enrich the field of cardiomyocyte proliferation research and enable the identification of new pathways that could modulate cardiomyocyte cell cycle progression.

## 9. The Premise to Cure Heart Failure and Current Obstacles

The above-mentioned studies suggest that cardiomyocyte cell cycle arrest may be reversed by direct induction of cell cycle factors or indirectly through the expression of transcription factors and miRNAs, the activation of signaling pathways, or metabolic modulation. In addition, most of these studies showed a significant improvement in cardiac function and reduction of scar size after MI in rodents. Hence, inducing the proliferation of existing cardiomyocytes could become a promising approach to cure heart failure in humans. However, clinical application of the induction of the proliferation of existing cardiomyocytes has many obstacles. First, the selective method of delivery and the duration of gene expression is a crucial obstacle due to the possibility of uncontrolled proliferation and the development of oncogenesis in other tissues. Second, the propagation of electrical stimulation in the myocardium is dependent on the cell junctions and it is crucial for synchronous contraction and force generation. It is still not clear if induction of the proliferation of existing cardiomyocytes will induce arrhythmogenic effects or negatively impact contractility, which was the case in one of the recent studies using miR-199a [64]. Third, the generated cardiomyocytes have to be homogeneous and have mature structural and functional properties. That is essential to avoid the possibility of an unnecessary increase in cardiac mass and the development of diastolic heart failure [64,112,130,131]. A recent preprint tried to address some of these limitations and demonstrated that transient and cardiomyocyte expression of cell cycle induction is a promising tool to safely induce cardiomyocyte proliferation, with no obvious toxicity in rodents and large animals [36].

## Figures and Tables

**Table 1 ijms-22-07720-t001:** Direct expression of cell cycle genes induces cardiomyocyte cell cycle entry.

Cell Cycle Gene	Studied Species	Highlights	References
Cyclin D1	Adult transgenic mice	Sustained DNA synthesis No evidence of complete cytokinesis	[26,28]
Cyclin D2	Adult transgenic mice	Observed DNA synthesis and reduced scar size after permanent coronary ligation in mice No evidence of complete cytokinesis	[13]
Cyclin G1	Mice hearts	Cyclin G1 expression stimulated DNA synthesis but arrested cytokinesis	[27]
Cyclin A2	Transgenic mice Rats, pigs	Modest cell proliferation in vitro but limited in vivo	[29,30,31,32]
CDK4–cyclin D1	Sprague–Dawley rats	Construction of a nuclear-localized cyclin D1 (D1NLS) to induce DNA synthesis	[33]
Cyclin B1–CDC2AF	Fetal, neonatal, and adult cardiomyocytes isolated from Wistar rat cardiomyocytes	Observation of increased cell numbers 72 h post-transfection by direct cell counting	[34]
CDK1, CDK4, cyclin B1, and cyclin D1	Post-mitotic mouse, rat, and human cardiomyocytes	Could induce stable cytokinesis in 15–20% of the adult cardiomyocytes expressing the four factors Self-limited through proteasome-mediated degradation of the protein products	[35,36]
P21 deacetylation by Sirt-1	Mice hearts	Sirt-1 deacetylated P21 and promoted its ubiquitination to release its inhibitory effect on the cell cycle	[37]
Cyclin-dependent kinase inhibitors (CKIs): P21, P27, and P57	SWISS CD1 mice Wistar rats	Knockdown of CKI using siRNA resulted in S phase induction. The subpopulation of cardiomyocytes progressed beyond karyokinesis	[23]
p38 MAP kinase inhibition	Rat cardiomyocytes	P38 knockout was associated with a 92.3% increase in neonatal cardiomyocyte mitosis and could promote cytokinesis in adult cardiomyocytes	[24]

**Table 2 ijms-22-07720-t002:** Transcription factors that regulate cardiomyocyte cell cycle entry.

Transcription Factor	Studied Species	Highlights	References
E2F	siRNA-mediated knockdown in neonatal mouse and rat cardiomyocytes	E2F4 was required during the G2/M phase through induction of cyclin A and cyclin E	[39,40]
	Adenovirus expression in mouse heart	E2F2 overexpression significantly increased the number of BrdU, PHH3, and Aurora kinase-positive cardiomyocytes	
Meis1	Knockout mice	Meis1 KO in mice extended the postnatal proliferation capacity of cardiomyocytes to 14 days Meis1 KO showed an increase in the cell cycle entry markers in cardiomyocytes	[51]
Tbx20	Induced specific overexpression in an adult mouse	Tbx20 overexpression promoted cardiomyocyte cell cycle entry Tbx20 overexpression improved cardiac function and scar size	[54]
Gata4	GATA4 KO mice Adenovirus expression in adult mice	GATA 4 KO mice showed a larger scar size, reduced angiogenesis, and cardiomyocyte cell cycle entry after P0 cryoinjury GATA4 overexpression promoted cardiomyocyte cell cycle entry	[59]
HOX13	Knockout mice	Hox13 KO increased cardiomyocyte cell cycle entry	[60]

**Table 3 ijms-22-07720-t003:** microRNAs that are involved in cardiomyocyte cell cycle entry.

miRNA	Studied Species	Highlights	References
miR-302-367	Transgenic mice	miR-302-367 overexpression promoted cardiomyocyte proliferation and reduced scar size after MI	[68]
miR-499	P19CL6 cells	miR-499 regulated the proliferation and apoptosis of P19CL6 cells in the late stage of cardiac differentiation	[67]
miR-199a	AAV overexpressing mirRNA199a	miR-199a overexpression promoted cardiomyocyte cell cycle entry in mice, rats, and pigs and induced almost complete recovery of cardiac function parameters after MI Long-term expression of miR199a resulted in cardiac death	[64,69]
miR-128	KO mice	Neonatal cardiomyocyte KO miR-128 showed prolonged proliferation Adult KO mice showed improved scar size and cardiac function after MI	[65]
miR-294	AAV overexpression	miR-294 overexpression promoted cardiomyocyte cell cycle entry	[66]

**Table 4 ijms-22-07720-t004:** Signaling pathways that influence cardiomyocyte cell cycle entry.

Pathway	Studied Species	Highlights	References
PI3K-AKT	Isolated rat cardiomyocytes	Activating the PI3K-AKT signaling pathway showed a successful induction of mitosis and cytokinesis	[74]
Hippo signaling pathway	Neonatal and adult mice cardiomyocytes. Transgenic mice	Active version could program adult mice cardiomyocytes to a fetal heart program with high proliferation capacity and increased chromatin accessibility. Cardiac-specific deletion of YAP in mice was associated with impaired cardiac functions with ventricular wall fibrosis Overexpressing a constitutively active YAP showed enhanced proliferative capacity and increased PHH3 expression after LAD artery ligation in P7 day old neonatal mice	[81,82,83]
Notch signaling pathway	Zebrafish	Suppression of Notch signaling in zebrafish impaired their capacity to regenerate their heart after ventricular resection	[86]
Wnt/β-catenin	sFRP2 and GSK3 β knockout mice	Knockout of sFRP2 protein showed reduced fibrosis and improved cardiac functions after MI GSK3β knockout activated Wnt/β-catenin, which increased cyclin D and allowed the cardiomyocytes to proceed into the S phase.	[89,90,91]
Jak/Stat	Zebrafish Adult mice cardiomyocytes	Stat3 facilitates cell cycle re-entry Stat3 inhibition was associated with a limited ability of cardiomyocytes to proliferate and increased scarring in response to injury	[92,93]
Hedgehog (Hh)	Zebrafish Newts	Hh signaling stimulated the expression of cyclin D2 and cyclin E1 and inhibited P27 through Gli1–Mycn network interaction	[102]
Autonomic innervation	Zebrafish Mice Human pediatric patients with F4	Chemical sympathectomy inhibited cardiac regeneration after apical resection β-adrenergic receptor stimulation in patients with tetralogy of Fallot (F4) was linked to cardiomyocyte cell cycle arrest Inhibitions of cholinergic nerve functions diminished cardiomyocyte proliferation Cholinergic stimulation extended the postnatal cardiomyocyte regeneration window	[103,104,105]

**Table 5 ijms-22-07720-t005:** Metabolic changes modulate cardiomyocyte cell cycle entry.

Metabolic Factor	Gene Involved	Highlights	Reference
Hypoxia	HIF-alpha	Hypoxia promoted oxidative damage and cardiomyocyte division	[106,107]
Mitochondrial substrate utilization	PDK4	Reduced fatty acid utilization and increased glucose utilization promoted cardiomyocyte cell cycle entry	[109]
Glycolysis	Pkm2	Glycolysis regulated cardiomyocyte cell cycle entry through activation of B-catenin	[110,111]
Switch from oxidative phosphorylation to glycolysis and lactate fermentation	ErB2	Metabolic switch to glycolysis and lactate fermentation promoted cardiomyocyte cell cycle entry	[112,113]
Succinate	SDH	Inhibition of succinate dehydrogenase promoted cardiomyocyte cell cycle entry in neonatal and adult cardiomyocytes	[114]
Mevalonate anabolic pathway		Promoted cardiomyocyte cell cycle entry	[116]

## Data Availability

Not applicable.

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
