# Peer review of "Induced Cardiomyocyte Proliferation: A Promising Approach to Cure Heart Failure"

_ijms, 2021, doi:10.3390/ijms22147720_

Round 1

Reviewer 1 Report

Promoting cardiac repair following pathologic injury remains an important area of cardiac research. Cardiomyocyte proliferation has gained renewed interest as an approach for mitigating damage and restoring lost myocardium in damaged hearts. In this manuscript, the authors review studies addressing cardiomyocyte proliferation, underlying mechanisms and potential targets for therapeutic intervention.

Overall the manuscript is well written, explains current understanding in the field, and includes tables that nicely summarize each section. The main concern is that several publications important to the ongoing discussion surrounding cardiomyocyte cell cycle and/or proliferation need to be included. Additionally, claims of proliferation and repair are overstated in some sections where cell cycle re-entry or salvage of myocardium are also plausible explanations of experimental results.

Comments intended to improve the manuscript for publication in IJMS are as follows:

Use of the word proliferation in reference to adult mammalian cardiomyocytes must be used carefully. Cell cycle entry or upregulation of cell cycle markers is not concrete evidence of cytokinesis.

As acknowledged by the authors, adult mammalian cardiomyocytes are notoriously resistant to cell division. To make matters worse, cardiomyocytes undergo cellular processes such as bi (or multi) nucleation, endoreplication, polyploidization and hypertrophy, for example, that can stimulate upregulation of cell cycle markers without completion of cell division. As a consequence, the term proliferation is often misused in the literature with respect to postnatal and/or post-replicative cardiomyocyte cellular and molecular biology. Furthermore, some transgenic mouse models overexpressing cell cycle or pro-proliferative proteins in cardiomyocytes may exhibit cardiomyocyte hyperplasia due to extended proliferation/delayed terminal differentiation in the early postnatal period rather than cell division in adult cardiomyocytes.

More recent, detailed studies aimed at teasing apart these distinctions and defining cardiomyocyte cell division need to be included to capture this complexity.

Midbody Positioning and Distance Between Daughter Nuclei Enable Unequivocal Identification of Cardiomyocyte Cell Division in Mice.

Hesse M, Doengi M, Becker A, Kimura K, Voeltz N, Stein V, Fleischmann BK.

Circ Res. 2018 Oct 12;123(9):1039-1052. 

Cardiomyocyte binucleation is associated with aberrant mitotic microtubule distribution, mislocalization of RhoA and IQGAP3, as well as defective actomyosin ring anchorage and cleavage furrow ingression.

Leone M, Musa G, Engel FB.Cardiovasc Res. 2018 Jul 1;114(8):1115-1131.

Pseudo-bipolar spindle formation and cell division in postnatal binucleated cardiomyocytes.

Leone M, Engel FB.J Mol Cell Cardiol. 2019 Sep;134:69-73.

Advances in heart regeneration based on cardiomyocyte proliferation and regenerative potential of binucleated cardiomyocytes and polyploidization.

Leone M, Engel FB.Clin Sci (Lond). 2019 Jun 7;133(11):1229-1253.

Although controversial, this publication needs to be addressed:

Cardiomyocytes Replicate and their Numbers Increase in Young Hearts

Nawazish Naqvi 1, Reena Singh 2, Siiri E Iismaa 3, Ming Li 3, John W Calvert 4, David I K Martin 5, Richard P Harvey 2, Robert M Graham 6, Ahsan Husain 7

Discuss and include recent publications using the FUCCI reporter system to document cardiomyocyte cell cycle dynamics in vitro, during development and in adult cardiomyocytes:

For example:

Cardiomyocyte cell cycle dynamics and proliferation revealed through cardiac-specific transgenesis of fluorescent ubiquitinated cell cycle indicator (FUCCI).

Alvarez R Jr, Wang BJ, Quijada PJ, Avitabile D, Ho T, Shaitrit M, Chavarria M, Firouzi F, Ebeid D, Monsanto MM, Navarrete N, Moshref M, Siddiqi S, Broughton KM, Bailey BA, Gude NA, Sussman MA.J Mol Cell Cardiol. 2019 Feb;127:154-164.

In vivo analysis of cardiomyocyte proliferation during trabeculation.

Uribe V, Ramadass R, Dogra D, Rasouli SJ, Gunawan F, Nakajima H, Chiba A, Reischauer S, Mochizuki N, Stainier DYR.Development. 2018 Jul 30;145(14):dev164194.

Fluorescent indicators for continuous and lineage-specific reporting of cell-cycle phases in human pluripotent stem cells.

Chang Y, Hellwarth PB, Randolph LN, Sun Y, Xing Y, Zhu W, Lian XL, Bao X.Biotechnol Bioeng. 2020 Jul;117(7):2177-2186.

doi: 10.1002/bit.27352. Epub 2020 Apr 22.PMID: 32277708 

Live cell screening platform identifies PPARdelta as a regulator of cardiomyocyte proliferation and cardiac repair.

Magadum A, Ding Y, He L, Kim T, Vasudevarao MD, Long Q, Yang K, Wickramasinghe N, Renikunta HV, Dubois N, Weidinger G, Yang Q, Engel FB.

Discuss how PI3K/Akt and other signaling pathways mentioned in the manuscript act as survival pathways as well as agents proliferation, and how beneficial effects described post infarction may represent salvage rather than repair due to cardiomyocyte proliferation.

In the PI3K/Akt section, cite the following publications: 

Akt promotes increased cardiomyocyte cycling and expansion of the cardiac progenitor cell population.

Gude N, Muraski J, Rubio M, Kajstura J, Schaefer E, Anversa P, Sussman MA.Circ Res. 2006 Aug 18;99(4):381-8. doi: 10.1161/01.RES.0000236754.21499.1c. Epub 2006 Jul 13.PMID: 16840722

Myocardial AKT: the omnipresent nexus

Mark A Sussman 1, Mirko Völkers, Kimberlee Fischer, Brandi Bailey, Christopher T Cottage, Shabana Din, Natalie Gude, Daniele Avitabile, Roberto Alvarez, Balaji Sundararaman, Pearl Quijada, Matt Mason, Mathias H Konstandin, Amy Malhowski, Zhaokang Cheng, Mohsin Khan, Michael McGregor

In the section on Notch signaling, discuss and include more publications addressing signaling  between Notch and PI3K in adult mammalian heart.

The writing in Section 4 on MicroRNA needs proofreading for grammar and syntax.

Author Response

Reviewer 1:

Comment 1:
Overall the manuscript is well written, explains current understanding in the field, and includes tables that nicely summarize each section. The main concern is that several publications important to the ongoing discussion surrounding cardiomyocyte cell cycle and/or proliferation need to be included. Additionally, claims of proliferation and repair are overstated in some sections where cell cycle re-entry or salvage of myocardium are also plausible explanations of experimental results.

Response:

We would like to thank the reviewer for his constructive comment.

Comment 2:

Comments intended to improve the manuscript for publication in IJMS are as follows:

Use of the word proliferation in reference to adult mammalian cardiomyocytes must be used carefully. Cell cycle entry or upregulation of cell cycle markers is not concrete evidence of cytokinesis.

Response:

We fully agree with the reviewer, and we have adjusted the terminology of proliferation to cell cycle entry if there is no proof of cytokinesis provided by the authors and they depended on only cell cycle marker activation. However, if the authors provided an evidence for true cytokinesis such as the use of MADM mice or they referred to proliferation during neonatal stage so this is still considered as true proliferation.

Comment 3:

As acknowledged by the authors, adult mammalian cardiomyocytes are notoriously resistant to cell division. To make matters worse, cardiomyocytes undergo cellular processes such as bi (or multi) nucleation, endoreplication, polyploidization and hypertrophy, for example, that can stimulate upregulation of cell cycle markers without completion of cell division. As a consequence, the term proliferation is often misused in the literature with respect to postnatal and/or post-replicative cardiomyocyte cellular and molecular biology. Furthermore, some transgenic mouse models overexpressing cell cycle or pro-proliferative proteins in cardiomyocytes may exhibit cardiomyocyte hyperplasia due to extended proliferation/delayed terminal differentiation in the early postnatal period rather than cell division in adult cardiomyocytes.

More recent, detailed studies aimed at teasing apart these distinctions and defining cardiomyocyte cell division need to be included to capture this complexity.

Midbody Positioning and Distance Between Daughter Nuclei Enable Unequivocal Identification of Cardiomyocyte Cell Division in Mice.

Hesse M, Doengi M, Becker A, Kimura K, Voeltz N, Stein V, Fleischmann BK.

Circ Res. 2018 Oct 12;123(9):1039-1052. 

Cardiomyocyte binucleation is associated with aberrant mitotic microtubule distribution, mislocalization of RhoA and IQGAP3, as well as defective actomyosin ring anchorage and cleavage furrow ingression.

Leone M, Musa G, Engel FB.Cardiovasc Res. 2018 Jul 1;114(8):1115-1131.

 Pseudo-bipolar spindle formation and cell division in postnatal binucleated cardiomyocytes.

Leone M, Engel FB.J Mol Cell Cardiol. 2019 Sep;134:69-73.

Advances in heart regeneration based on cardiomyocyte proliferation and regenerative potential of binucleated cardiomyocytes and polyploidization.

Leone M, Engel FB.Clin Sci (Lond). 2019 Jun 7;133(11):1229-1253.

 Although controversial, this publication needs to be addressed:

Cardiomyocytes Replicate and their Numbers Increase in Young Hearts

Nawazish Naqvi 1, Reena Singh 2, Siiri E Iismaa 3, Ming Li 3, John W Calvert 4, David I K Martin 5, Richard P Harvey 2, Robert M Graham 6, Ahsan Husain 7

 Discuss and include recent publications using the FUCCI reporter system to document cardiomyocyte cell cycle dynamics in vitro, during development and in adult cardiomyocytes:

 For example:

Cardiomyocyte cell cycle dynamics and proliferation revealed through cardiac-specific transgenesis of fluorescent ubiquitinated cell cycle indicator (FUCCI).

Alvarez R Jr, Wang BJ, Quijada PJ, Avitabile D, Ho T, Shaitrit M, Chavarria M, Firouzi F, Ebeid D, Monsanto MM, Navarrete N, Moshref M, Siddiqi S, Broughton KM, Bailey BA, Gude NA, Sussman MA.J Mol Cell Cardiol. 2019 Feb;127:154-164.

In vivo analysis of cardiomyocyte proliferation during trabeculation.

Uribe V, Ramadass R, Dogra D, Rasouli SJ, Gunawan F, Nakajima H, Chiba A, Reischauer S, Mochizuki N, Stainier DYR.Development. 2018 Jul 30;145(14):dev164194.

Fluorescent indicators for continuous and lineage-specific reporting of cell-cycle phases in human pluripotent stem cells.

Chang Y, Hellwarth PB, Randolph LN, Sun Y, Xing Y, Zhu W, Lian XL, Bao X.Biotechnol Bioeng. 2020 Jul;117(7):2177-2186.

doi: 10.1002/bit.27352. Epub 2020 Apr 22.PMID: 32277708 

 Live cell screening platform identifies PPARdelta as a regulator of cardiomyocyte proliferation and cardiac repair.

Magadum A, Ding Y, He L, Kim T, Vasudevarao MD, Long Q, Yang K, Wickramasinghe N, Renikunta HV, Dubois N, Weidinger G, Yang Q, Engel FB.

Response:

We fully agree with the reviewer, however, this was out of the focus of this review. But to satisfy the reviewer comment, we have included a new section entitled “Recent genetic tools to track cardiomyocyte cell cycle” to summarize the recent efforts in developing genetic tools to track the cell cycle progression in cardiomyocytes. We have cited the highlighted articles by the reviewers in this section

Comment 4:

Discuss how PI3K/Akt and other signaling pathways mentioned in the manuscript act as survival pathways as well as agents proliferation, and how beneficial effects described post infarction may represent salvage rather than repair due to cardiomyocyte proliferation.

In the PI3K/Akt section, cite the following publications: 

Akt promotes increased cardiomyocyte cycling and expansion of the cardiac progenitor cell population.

Gude N, Muraski J, Rubio M, Kajstura J, Schaefer E, Anversa P, Sussman MA.Circ Res. 2006 Aug 18;99(4):381-8. doi: 10.1161/01.RES.0000236754.21499.1c. Epub 2006 Jul 13.PMID: 16840722

Myocardial AKT: the omnipresent nexus

Mark A Sussman 1, Mirko Völkers, Kimberlee Fischer, Brandi Bailey, Christopher T Cottage, Shabana Din, Natalie Gude, Daniele Avitabile, Roberto Alvarez, Balaji Sundararaman, Pearl Quijada, Matt Mason, Mathias H Konstandin, Amy Malhowski, Zhaokang Cheng, Mohsin Khan, Michael McGregor

Response:

We would like to thank the reviewer and we have discussed the effect of PI3K/AKT on cell survival and cited the highlighted reference.

Comment 5:

In the section on Notch signaling, discuss and include more publications addressing signaling  between Notch and PI3K in adult mammalian heart.

Response:

As requested by the reviewer, we discussed the correlation between Notch and PI3K/AKT signaling.

Comment 6:

The writing in Section 4 on MicroRNA needs proofreading for grammar and syntax.

Response:

We revised the grammar and syntax in this section.

Reviewer 2 Report

This is a comprehensive review article on the promotion of myocardial proliferation, which is particularly important in myocardial regenerative medicine, myocardial infarction, and other related medical treatments. The author discussed direct cell cycle activation and cell cycle-associated transcriptional factors, miRNA, signal molecules and metabolites, etc., and made a review of relevant literature over the past few decades. I think that this article is ready to be published and it will be valuable not only in theory but also in practice.

I have two questions and one suggestion as follows.

Two questions:

  1. On page 4 of 14, line 173, should the “PI3/AKT” be revised as “PI3K/AKT”?
  2. On page 5 of 14, line 209, the article mentions “miR-128 KO mice showed improved scar size and cardiac function after MI”. Does it mean that this KO increases mesenchymal fibroblast but not myocardial proliferation?

One suggestion:

Whether to consider including the following points of discussion (not necessary): "the adverse effects of cell cycle activation in adult CM or iPS-CM".

Author Response

Reviewer 2:

Comment 1:

This is a comprehensive review article on the promotion of myocardial proliferation, which is particularly important in myocardial regenerative medicine, myocardial infarction, and other related medical treatments. The author discussed direct cell cycle activation and cell cycle-associated transcriptional factors, miRNA, signal molecules and metabolites, etc., and made a review of relevant literature over the past few decades. I think that this article is ready to be published and it will be valuable not only in theory but also in practice.

Response:

We would like to thank the reviewer for his appreciation of our efforts.

Comment 2:

On page 4 of 14, line 173, should the “PI3/AKT” be revised as “PI3K/AKT”?

Response:

We would like to thank the reviewer for identifying this typo which was corrected in the revised manuscript.

Comment 3:

On page 5 of 14, line 209, the article mentions “miR-128 KO mice showed improved scar size and cardiac function after MI”. Does it mean that this KO increases mesenchymal fibroblast but not myocardial proliferation?

Response:

According to this study the authors suggested that the reduction in scar size was attributed to the dedifferentiation of the adult cardiomyocytes and their ability to proliferate and hence preventing the development of fibrosis. We included this in our revised version.

Comment 4:

Whether to consider including the following points of discussion (not necessary): "the adverse effects of cell cycle activation in adult CM or iPS-CM".

Response:

We fully agree with the reviewer, and this was already included under the section “The premise to cure heart failure” in the first submission where we highlighted the several limitations for such therapeutic approach such as induction of arrhythmia and production of immature